∂ | **Open Peer Review** | Bacteriology | Research Article

# A novel vaccine strategy using quick and easy conversion of bacterial pathogens to unnatural amino acid-auxotrophic suicide derivatives

Yuya Nagasawa,[1] Momoko Nakayama,[2] Yusuke Kato,[3] Yohsuke Ogawa,[1] Swarmistha Devi Aribam,[2] Yusaku Tsugami,[1] Taketoshi Iwata,[2] Osamu Mikami,[1] Aoi Sugiyama,[1] Megumi Onishi,[1] Tomohito Hayashi,[1] Masahiro Eguchi[2]

**ABSTRACT**  We propose a novel strategy for quick and easy preparation of suicide live vaccine candidates against bacterial pathogens. This method requires only the transformation of one or more plasmids carrying genes encoding for two types of biological devices, an unnatural amino acid (uAA) incorporation system and toxin-anti-toxin systems in which translation of the antitoxins requires the uAA incorporation. *Escherichia coli* BL21-AI laboratory strains carrying the plasmids were viable in the presence of the uAA, whereas the free toxins killed these strains after the removal of the uAA. The survival time after uAA removal could be controlled by the choice of the uAA incorporation system and toxin-antitoxin systems. Multilayered toxin-antitoxin systems suppressed escape frequency to less than 1 escape per $10^9$ generations in the best case. This conditional suicide system also worked in *Salmonella enterica* and *E. coli* clinical isolates. The *S. enterica* vaccine strains were attenuated with a $>10^5$ fold lethal dose. Serum IgG response and protection against the parental pathogenic strain were confirmed. In addition, the live *E. coli* vaccine strain was significantly more immunogenic and provided greater protection than a formalin-inactivated vaccine. The live *E. coli* vaccine was not detected after inoculation, presumably because the uAA is not present in the host animals or the natural environment. These results suggest that this strategy provides a novel way to rapidly produce safe and highly immunogenic live bacterial vaccine candidates.

**IMPORTANCE**  Live vaccines are the oldest vaccines with a history of more than 200 years. Due to their strong immunogenicity, live vaccines are still an important category of vaccines today. However, the development of live vaccines has been challenging due to the difficulties in achieving a balance between safety and immunogenicity. In recent decades, the frequent emergence of various new and old pathogens at risk of causing pandemics has highlighted the need for rapid vaccine development processes. We have pioneered the use of uAAs to control gene expression and to conditionally kill host bacteria as a biological containment system. This report proposes a quick and easy conversion of bacterial pathogens into live vaccine candidates using this containment system. The balance between safety and immunogenicity can be modulated by the selection of the genetic devices used. Moreover, the uAA-auxotrophy can prevent the vaccine from infecting other individuals or establishing the environment.

**KEYWORDS**  vaccine development strategy, unnatural amino acids, toxin-antitoxin systems, bacterial pathogens

E merging and re-emerging infectious diseases are among the most serious threats to public health (1). High population densities due to urban concentration and

Address correspondence to Yusuke Kato, kato@affrc.go.jp, Tomohito Hayashi, hayashi-tomohito@zenoaq.jp, or Masahiro Eguchi, egumaro@affrc.go.jp.

Yuya Nagasawa, Momoko Nakayama, and Yusuke Kato contributed equally to this article. The order of co-first authors has been determined following prior agreement.

The authors declare no conflict of interest.

See the funding table on p. 17.

increased contact between people around the world due to widespread air travel have greatly facilitated the global spread of pathogens (2, 3). The emergence of pandemics in recent decades has highlighted the urgent need to develop strategies to control these infectious diseases (4). Vaccination is a promising strategy. Currently, the average development time of conventional vaccines from the preclinical stage is more than 10 years (5). However, the ongoing story of the fight against the global COVID-19 pandemic indicates the need for a much more rapid vaccine development strategy (6). Bacterial pathogens, including the ever-emerging antibiotic-resistant bacteria, pose a threat, as do viruses, including SARS coronaviruses, dengue viruses, and pandemic influenza viruses (7).

Today, new vaccine technologies such as virus-like particle vaccines, nanoparticle vaccines, DNA/RNA vaccines, and rational vaccine design are being developed in addition to traditional inactivated vaccines, live attenuated vaccines, viral vector vaccines, and subunit vaccines (8, 9). Among these, live attenuated bacterial vaccines are the oldest in use and still one of the leading vaccine categories (10, 11). Because live vaccines mimic natural infection almost perfectly, they can induce a broader range of immune responses in both humoral and cellular immunity. However, the development of live vaccines presents a unique dilemma (12). Namely, inadequate attenuation would result in high virulence and compromised safety while excessive attenuation would compromise immunogenicity. This trade-off makes it difficult to obtain a live vaccine strain that balances immunogenicity and safety.

Here, we propose a quick and easy method to generate live vaccine candidates with balanced safety and immunogenicity using a conditional suicide system that can be achieved "additively" by plasmid transformation alone (Fig. 1). This method is an application of our previously reported technique (13). This technique renders bacteria auxotrophic for an unnatural amino acid (uAA) that does not exist in the natural environment or living organisms. Briefly, one or more toxin-antitoxin gene pairs are transformed into pathogens using plasmids. We selected type II, type IV, or type V toxin-antitoxins whose antitoxin genes encode proteins (14). A UAG stop codon(s) is inserted next to the translation initiation codon of the antitoxin genes. To incorporate the uAA at the UAG stop codon, the uAA-specific aminoacyl-tRNA synthetase and its cognate tRNA$_{CUA}$ gene are also transformed (15). In the absence of uAA, no functional antitoxin is produced due to translation termination at the inserted UAG stop codon, resulting in the killing of the host bacteria by the constitutively expressed toxin (16, 17). By contrast, bacteria survive and proliferate in the presence of uAA because the toxin is neutralized by the functional antitoxin produced by the readthrough of the UAG stop codon. Thus, the pathogen can be converted into a conditional suicide strain that can only survive in the presence of the uAA by simply transforming these plasmids. For use as a live vaccine, the uAA-auxotrophic strains are first cultured *in vitro* in the presence of uAA to obtain the desired dose and then administered to humans or animals. Since the uAA has accumulated in the vaccine cells just after the supply is cut off, they survive for a while and could be immunogenic like the parental pathogens (13). Thereafter, the vaccine strain is expected to die over time and not cause disease.

Unlike conventional live-attenuated vaccines, the uAA-auxotrophic strains are "suicide vaccines" that ensure safety by conditional suicide of the inoculated bacteria. A *Listeria monocytogenes* suicide vaccine using a kill switch that is activated only in the host cell's cytosol has been successfully constructed (18). In *Salmonella enterica*, an arabinose-dependent suicide vaccine was established (19). Our method is expected to have the following unique features. First, candidate vaccine strains can be generated rapidly. In modern live vaccine development, the induction of random or targeted mutations in genes of virulence-related and/or metabolic pathways is mainly used to attenuate wild-type pathogens (20). In this well-known method, vaccine strains are generated by "subtracting" genetic elements from the pathogen. This requires modification of the native genetic information of the pathogen, which is time-consuming and labor-intensive. Our method is quick and easy because it is an "additive" method that only

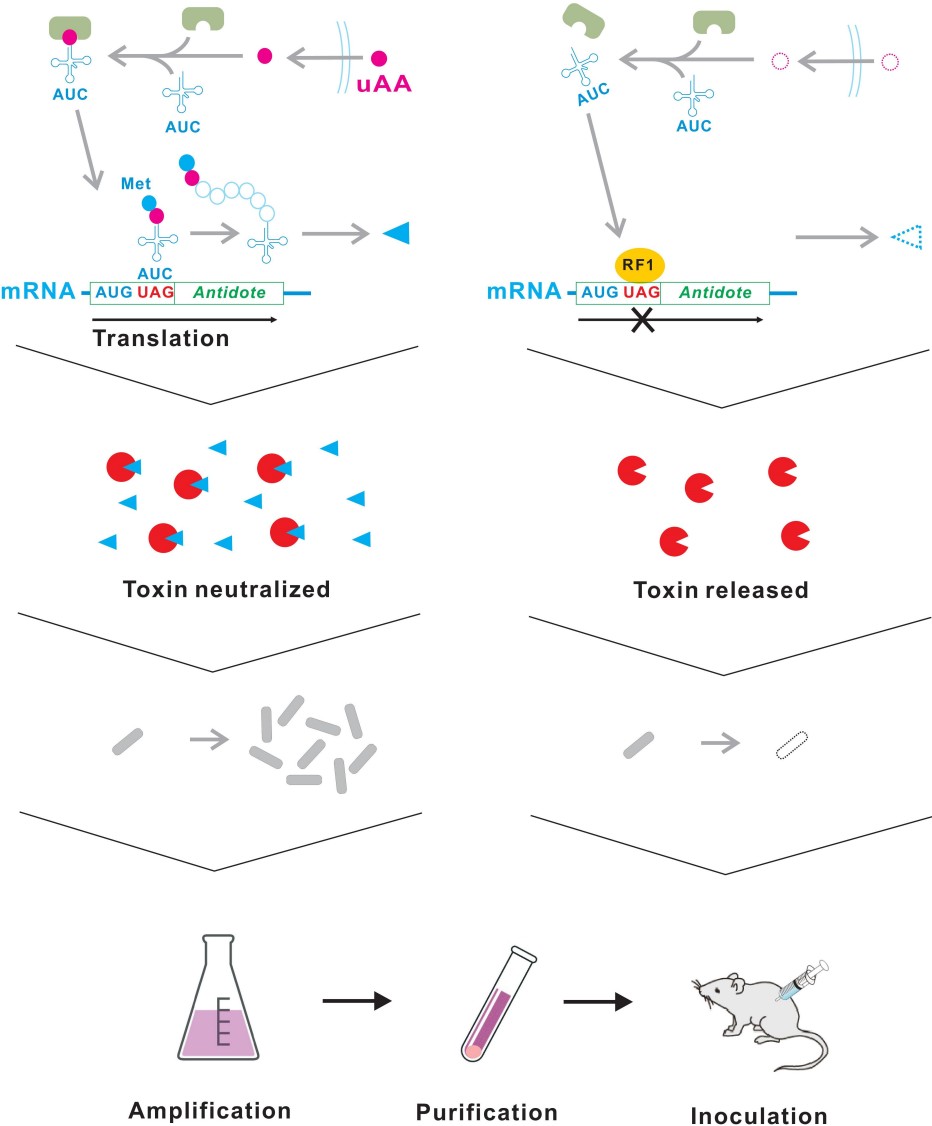

**FIG 1** Schematic diagram of the uAA-auxotrophic suicide bacterial vaccine. Vaccine strains can grow in the presence of uAA. After purification from the culture medium, the bacterial vaccines are inoculated into humans or animals. After administration, the bacterial vaccine gradually dies due to depletion of the intracellular uAA. RF1, peptide release factor 1.

transforms the plasmids. This simple method should facilitate screening to select live vaccine strains with sufficient safety and immunogenicity. Second, the attenuation can be controlled by the user, allowing a balance between safety and immunogenicity (12, 21). Since the native genetic information of the parental pathogen is not altered, the immunogenicity should be maintained as original during the viability of the vaccine candidates. The duration of viability for the vaccine candidates would depend on the choice of toxin-antitoxins, uAA incorporation system, and uAAs, suggesting that users can adjust the balance by adjusting these factors. Third, environmental risks are minimized. Live vaccines must be prevented from infecting unvaccinated individuals or spreading into the environment because of problems such as reversion to virulence, high susceptibility of immunocompromised individuals, and unintended immunization (22, 23). Biological containment, in which the target organism is genetically programmed to survive only under the control of the users and to die in the natural environment, is desired to prevent the spread of live vaccines used in the open environment (24–27). Many biological containments using auxotrophy for natural nutrients or conditional

toxin gene expression regulated by natural substances have often failed due to the presence of unexpected resources such as secretions and cell corpses (28, 29). Therefore, auxotrophy for substances not present in the natural environment is an ideal biological containment (13, 30). In our method, live vaccine candidates must be supplied with a uAA to survive. The uAAs are not present in the natural environment or living organisms, indicating that a high level of environmental safety should be ensured.

To demonstrate our idea, we constructed the suicidal *E. coli* laboratory strain BL21-AI auxotrophic for a uAA using the transformation of the conditionally toxic plasmids. Various combinations of the uAA incorporation systems and the toxin-antitoxin systems were examined to achieve the requirements necessary for vaccine construction, including low escape frequency and control of survival time after uAA removal. In addition, the resulting conditional suicide system was applied to two species of bacterial pathogens, *S. enterica* and *E. coli* clinical isolates, to evaluate vaccine safety and immunogenicity. *S. enterica* is an intracellular parasitic human and zoonotic pathogen (31). *E. coli* is an important intestinal commensal bacterium of vertebrates, including strains that cause a variety of gastrointestinal and extraintestinal diseases in humans and livestock (32). Vaccination is a promising way to prevent both *S. enterica* and *E. coli* infections (33, 34). Because *S. enterica* and *E. coli* are closely related phylogenetically (35), several plasmids replicate in both bacteria, including pBR322 and pACYC184, which harbor the pMB1 and p15A origin of replication, respectively (36). Thus, the *E. coli* BL21-AI conditional suicide systems were transferred intact to both *S. enterica* and *E. coli* clinical isolates. The resulting strains were inoculated into mice for vaccine evaluation.

## RESULTS

### Development of uAA-regulated conditionally toxic plasmids

We first developed uAA-regulated conditionally toxic plasmids using the *E. coli* laboratory strain BL21-AI (37). Previously, we constructed the BL21-AI derivative, BL21-AI(IY-o), auxotrophic for the uAA 3-iodo-$_L$-tyrosine (IY) by plasmid transformation (13) (Fig. 2). BL21-AI(IY-o) carries two plasmids, plasmid B-1 which contains the IY-incorporation system consisting of the IY-specific aminoacyl-tRNA synthetase and its cognate tRNA$_{CUA}$, and plasmid C1 which contains the toxin-antitoxin gene pair *colE3-immE3* (38) (Fig. 2; Fig. S1). The *immE3* contains a UAG-stop codon next to the translation start codon. BL21-AI(IY-o) died in the absence of IY but more than $10^3$ escapers/$10^8$ generations were detected, indicating that the escape frequency is $>1 \times 10^{-5}$ (13). This escape frequency is too high compared to the present standard for environmental safeguards set by the National Institutes of Health (NIH), $<1 \times 10^{-8}$ (39).

A lower escape frequency reduces the risk of pathogenesis and increases the upper limit of safe dosage, as well as minimizing the environmental risk. Therefore, we modified the plasmids to reduce the escape frequency. Even in the absence of IY, the IY incorporation system translates UAG stop codon inserted mRNAs at 6%–20% of the protein production in the presence of IY, presumably due to the misincorporation of natural amino acids such as Tyr (16, 40, 41). The high-level leakage production of ImmE3 could reduce ColE3 toxicity and increase the escape frequency. A positive-feedback genetic circuit in plasmid B-2 suppresses the leakage production to <1% (42), resulting in a 0.1-fold reduction in escape frequency in BL21-AI(IYm-o) strain (Fig. 2; Fig. S1 and S2). In addition, plasmid A-1, which contains a $N^\varepsilon$-benzyloxycarbonyl-$_L$-lysine (ZK) incorporation system consisting of the modified *Methanosarcina mazei* pyrrolysyl-tRNA synthetase specific for ZK and its cognate tRNA$_{CUA}$ whose leakage production level is only 2%, was used in BL21-AI(ZK-o) instead of plasmid B-1 or B-2 (42, 43). The escape frequency in BL21-AI(ZK-o) was further reduced to 0.1-fold of that in BL21-AI(IYm-o). Hence, we used the ZK incorporation system to construct strains with lower escape frequency.

Next, the effects of the toxin-antitoxin systems Kid-Kis and CcdB-CcdA were tested in addition to ColE3-ImmE3 (44, 45). Instead of *colE3-immE3, kid-kis,* and *ccdB-ccdA* were

### E. coli BL21-AI

| Strain name | Plasmid A | | | Plasmid B | Plasmid C | | Escape frequency (per $10^8$ generation) |
| --- | --- | --- | --- | --- | --- | --- | --- |
| | ZK | kid/1a-kis | ccdB/1a-ccdA | IY | colE3/1a-immE3 | ccdB/1a-ccdA | |
| BL21-AI(IY-o) | | | | B-1 | C-1 | | 1414 |
| BL21-AI(IYm-o) | | | | B-2 (p-loop) | C-1 | | 182 |
| BL21-AI(ZK-o) | A-1 | | | | C-1 | | 18.1 |
| BL21-AI(ZK-k) | A-2 | ● | | | | | 10.1 |
| BL21-AI(ZK-d) | A-1 | | | | | C-2 | 4.0 |
| BL21-AI(ZK-ko) | A-2 | ● | | | C-1 | | 1.37 |
| BL21-AI(ZK-kd) | A-3 | ● | ● | | | | <0.67 |
| BL21-AI(ZK-kdo) | A-3 | ● | ● | | C-1 | | <0.09 |

### N61

| Strain name | ZK | kid/1a-kis | ccdB/1a-ccdA | IY | colE3/1a-immE3 | ccdB/1a-ccdA | Escape frequency |
| --- | --- | --- | --- | --- | --- | --- | --- |
| N61(ZK-kdo) | A-3 | ● | ● | | C-1 | | 0.23 |

### H126

| Strain name | ZK | kid/1a-kis | ccdB/1a-ccdA | IY | colE3/1a-immE3 | ccdB/1a-ccdA | Escape frequency |
| --- | --- | --- | --- | --- | --- | --- | --- |
| H126(ZK-kdo) | A-3 | ● | ● | | C-1 | | 1.09 |

### S. enterica χ3306

| Strain name | ZK | kid/1a-kis | ccdB/1a-ccdA | IY | colE3/1a-immE3 | ccdB/1a-ccdA | Escape frequency |
| --- | --- | --- | --- | --- | --- | --- | --- |
| χ3306(ZK-k) | A-2 | ● | | | | | 4.58 |
| χ3306(ZK-kd) | A-3 | ● | ● | | | | 7.48 |

FIG 2 Escape frequency. Plasmids containing various uAA incorporation systems and toxin-antitoxin genes were transfected. Escape frequency in the absence of uAA was evaluated using a fluctuation assay. Values of the escape frequency are presented as escapers per $10^8$ generations. An inequality sign shows less than the indicated detection limit. ZK and IY, the ZK and IY incorporation system, respectively. A-1 to C-2, plasmid names shown in Fig. S1.

used in BL21-AI(ZK-k) and BL21-AI(ZK-d), respectively. Both BL21-AI(ZK-k) and BL21-AI(ZK-d) died in the absence of ZK, with an escape frequency comparable to that in BL21-AI(ZK-o) (Fig. 2; Fig. S1 and S2). The escape frequency remained at $>1 \times 10^{-8}$.

Multi-layered containment systems, in which two or more kill switches are used together, have been reported to show a lower escape frequency (46, 47). We, therefore, constructed strains carrying two or three toxin-antitoxin gene pairs. BL21-AI(ZK-ko) carrying both colE3-immE3 and kid-kis showed a lower escape frequency than strains carrying either the toxin-antitoxin gene pair. The escape frequency in BL21-AI(ZK-kd) carrying both kid-kis and ccdB-ccdA reached $<0.67 \times 10^{-8}$. Moreover, BL21-AI(ZK-kdo) carrying three toxin-antitoxin gene pairs, kid-kis, ccdB-ccdA, and colE3-immE3, recorded the lowest escape frequency at $<0.9 \times 10^{-9}$. The escape frequencies in BL21-AI(ZK-kd) and BL21-AI(ZK-kdo) qualified the current NIH standard.

Changes in survival after the uAA removal were evaluated because the change may primarily affect how long the viable suicidal vaccine persists in the inoculated human/animal. BL21-AI(IY-o) continued to grow for at least 1 h after IY removal presumably due to intracellular accumulation but then died rapidly with a half-life of approximately 50 min (13). It took around 6 h to decrease to 10% of the initial inoculation ($T_{0.1}$). The growth rate may be another important factor determining the initial proliferation immediately after the uAA removal and the intracellular uAA consumption. BL21-AI(IY-o) grew at 0.63 times the rate of a vector control strain (13). Growth of BL21-AI(ZK-o) was only observed within 15–30 min, and $T_{0.1}$ was 3–4 h (Fig. 3A). In BL21-AI(ZK-kd), the growth period and $T_{0.1}$ were largely extended to 2 h and 4–5 h, respectively. Curiously, BL21-AI(ZK-kdo) had a similar growth period and $T_{0.1}$ to those of BL21-AI(ZK-o), 15–30 min and 2–2.5 h, respectively. The growth rates of BL21-AI(ZK-o), BL21-AI(ZK-kd), and

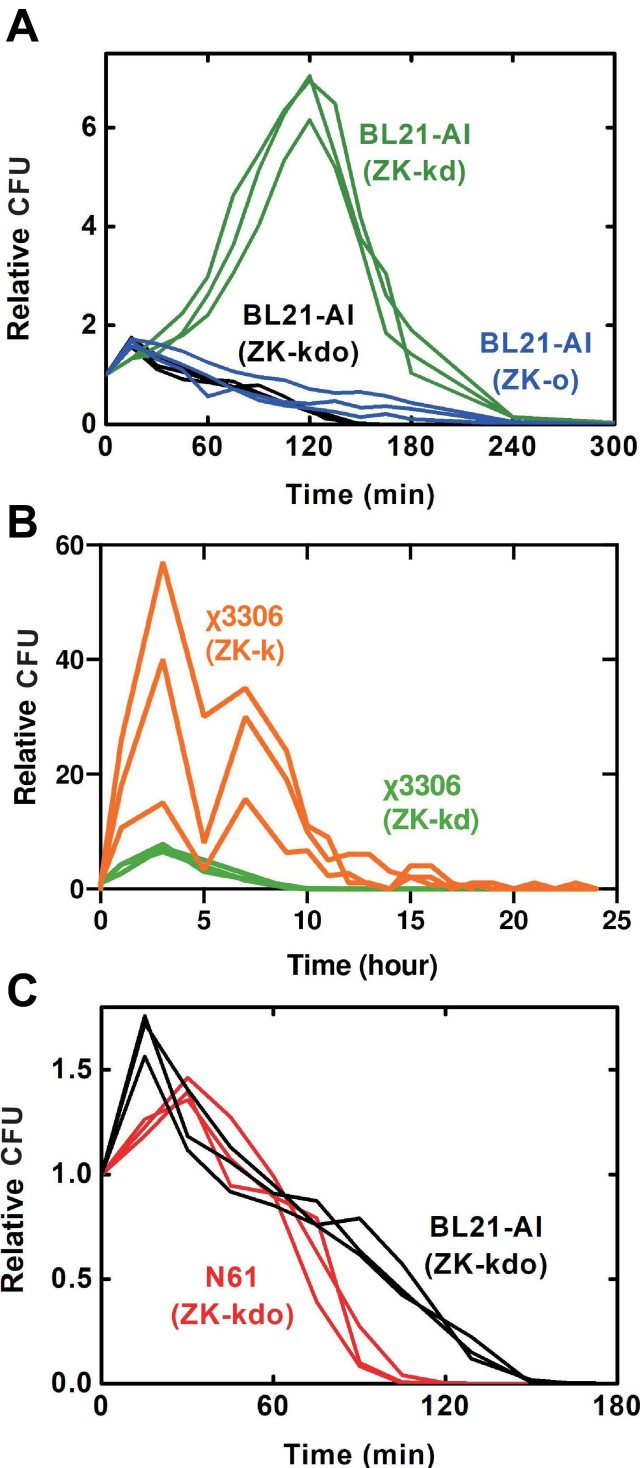

**FIG 3** Viability after ZK removal. Some ZK-auxotrophic vaccine candidates were tested in the logarithmic growth phase. After the exchange with a ZK-free medium, viable bacterial cells were counted over time. Values are relative to T0. (A) BL21-AI strains carrying various combinations of toxin-antitoxins. (B) *S. enterica* χ3306 (ZK-k) and χ3306 (ZK-kd) carrying A-2 and A-3, respectively. (C) Difference between BL21-AI and N-61 suicidal strains. Both strains carry A-3 and C-1 plasmids containing *kid-kis, ccdB-ccdA,* and *colE3-immE3*.

BL21-AI(ZK-kdo) decreased to 0.89, 0.68, and 0.72 of the vector control strains, respectively, similar to BL21-AI(IY-o) (Table S1).

## Application to *S. enterica*

We first constructed *S. enterica* suicide vaccine candidates using the uAA-regulated conditionally toxic plasmids. The plasmids developed using BL21-AI were applied for *S. enterica* without any modifications. A highly pathogenic strain against mice, χ3306, was used as a host (48). Plasmid A-2 or A-3 was successfully transformed into χ3306. The resulting strains were designated as χ3306 (ZK-k) and χ3306 (ZK-kd), respectively. Both χ3306 (ZK-k) and χ3306 (ZK-kd) were ZK-auxotrophic (Fig. S3). The escape frequency was slightly higher than the NIH standard (Fig. 2). To further reduce the escape frequency, we attempted to transform plasmid C-1 into χ3306 (ZK-k) and χ3306 (ZK-kd) but no transformants were obtained. Changes in survival after the uAA removal for χ3306 (ZK-k) and χ3306 (ZK-kd) were evaluated. The survival period of χ3306 (ZK-k) and χ3306 (ZK-kd) were longer than those of BL21-AI strains (Fig. 3B). In χ3306 (ZK-k), the growth periods and $T_{0.1}$ were largely prolonged to 3–7 h and 19–24 h, respectively. Growth periods and $T_{0.1}$ of χ3306 (ZK-kd) were 3 h and 10 h, respectively. The shape of the survival curve of χ3306 (ZK-kd), however, was similar to that of BL21-AI(ZK-kd). The growth rate of χ3306 (ZK-k) and χ3306 (ZK-kd) decreased to 0.62 and 0.93 of the vector control strains, respectively (Table S1).

To test the vaccine safety, χ3306 (ZK-k) and χ3306 (ZK-kd) were administered intravenously to BALB/c mice at $1 \times 10^5$ CFU/mouse. Although all mice inoculated with the wild-type χ3306 were dead in 4 days after inoculation, all mice administrated with χ3306 (ZK-k) or χ3306 (ZK-kd) survived (Fig. 4A). In addition, mice survived after intravenous administration of χ3306 (ZK-k) or χ3306 (ZK-kd) at $1 \times 10^6$ CFU/mouse.

After fourth-dose vaccination, single-dose administration at $1 \times 10^3$ CFU/mouse following triple-dose administration at $1 \times 10^4$ CFU/mouse for χ3306 (ZK-k) or χ3306 (ZK-kd), the immunized mice were challenged with intraperitoneal injection of the wild-type χ3306 at $1 \times 10$ CFU/mouse (Fig. 4B). We evaluated the induction of anti-*S. enterica* lipopolysaccharide (LPS) IgG after the intraperitoneal administration with χ3306 (ZK-k) and χ3306 (ZK-kd) (Fig. 4C). As shown in Fig. 4B and C, χ3306 (ZK-kd) immunized mice showed significantly higher anti-LPS IgG compared to controls at more than 28 days after immunization ($P = 0.007$). On the other hand, anti-LPS antibody levels in χ3306 (ZK-k) immunized mice did not differ significantly from controls even 56 days after immunization, but a tendency to increase induction of the anti-LPS IgG was observed.

Both χ3306 (ZK-k) and χ3306 (ZK-kd) vaccinated mice had 20% and 40% survival, respectively, at 35 days after the challenge. By contrast, all unvaccinated mice died by 12 days, suggesting that the vaccination with χ3306 (ZK-k) and χ3306 (ZK-kd) partially protected the mice against the wild-type χ3306 challenge (Fig. 4D).

To determine the efficacy of protection against oral infections, we also performed an oral challenge test (Fig. 4E). Mice were intravenously vaccinated with triple-dose administration at $1 \times 10^5$ CFU/mouse of χ3306 (ZK-k) or χ3306 (ZK-kd). When total IgG titers were assayed, production was observed after two inoculations (Fig. 4F). χ3306 (ZK-kd) and χ3306 (ZK-k) induced significantly more anti-LPS IgG than control at day 40 post-immunization ($P = 0.017$ and $P = 0.0005$, respectively). Immunized mice were challenged orally with $1.5 \times 10^5$ CFU/mouse of wild-type χ3306. All non-immunized mice died until 15 days after the challenge. Mice immunized with χ3306 (ZK-k) or χ3306 (ZK-kd) survived for 42 days (60% and 40%, respectively) (Fig. 4G).

Conclusively, the ZK-auxotrophic suicidal *S. enterica* vaccine is less virulent and immunogenic enough to induce antibody production and confer protection against the wild-type parent strain. Therefore, the highly virulent strain χ3306 has been successfully transformed into a live vaccine.

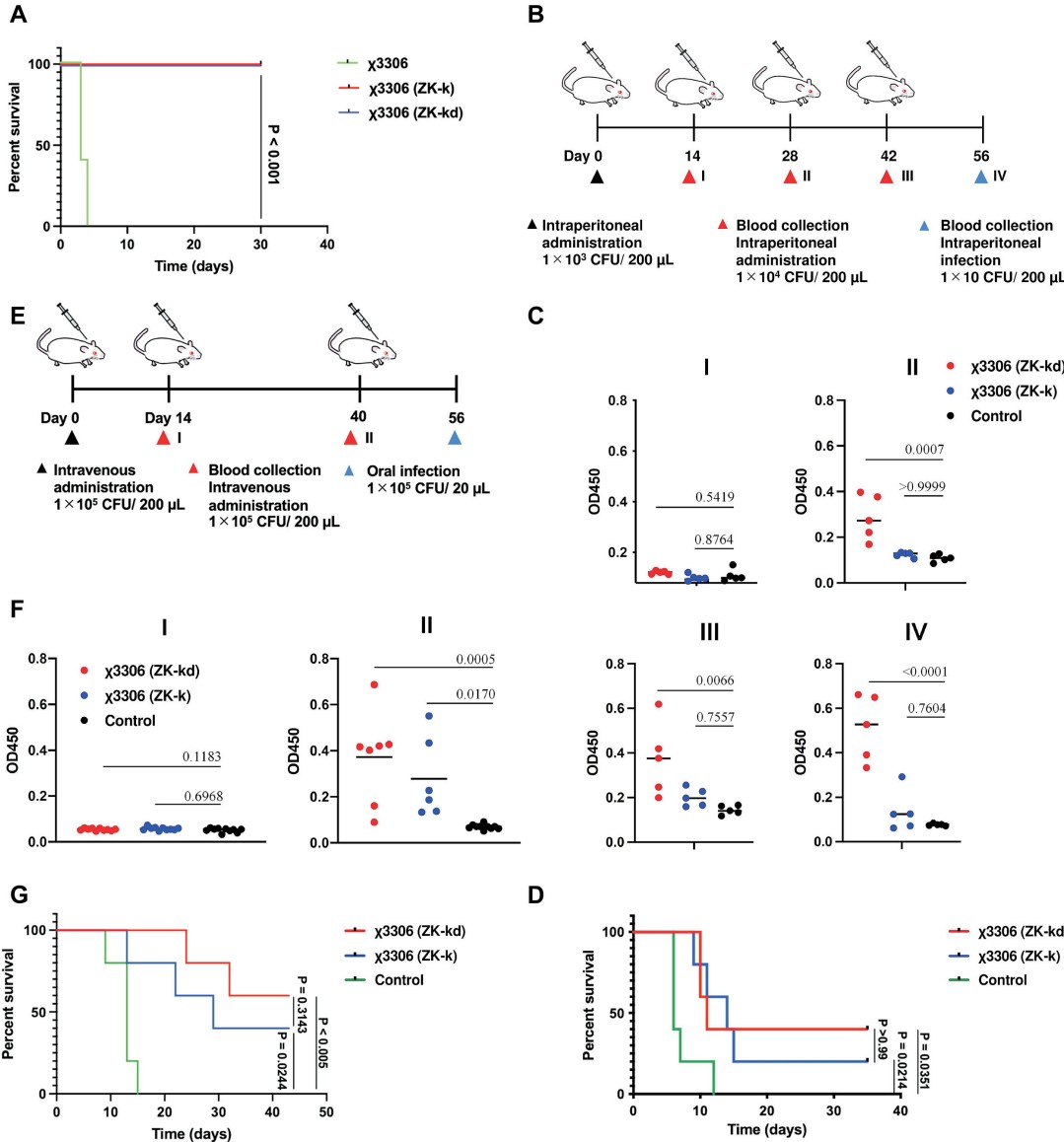

**FIG 4** Application to *S. enterica*. (A) Survival curves of mice intraperitoneally injected with *Salmonella* χ3306 (ZK-k) and χ3306 (ZK-kd), or wild-type χ3306 at the dose of $1 \times 10^5$ CFU/mouse (*n* = 5). (B) Experimental design to assess the systemic humoral response and protection against *Salmonella* infection. BALB/c mice were immunized intraperitoneally with χ3306 (ZK-k) and χ3306 (ZK-kd). After 56 days, BALB/c mice were intraperitoneally infected. (C) Antibody titers of total IgG against LPS. I, II, III, and IV indicate days 14, 28, 42 and 56, respectively. (D) Survival curves of *Salmonella* χ3306 (ZK-k) and χ3306 (ZK-kd) immunized mice intraperitoneally infected with χ3306 (wild type) (*n* = 5). (E) Experimental design to assess protection against *Salmonella* oral infection by χ3306 (ZK-k) and χ3306 (ZK-kd) immunized mice. BALB/c mice were immunized intravenously with χ3306(ZK-k) and χ3306(ZK-kd), and 56 days later, BALB/c mice were orally infected. (F) Total IgG antibody titers against LPS. I and II indicate days 14 and 40, respectively. (G) Survival curves of *Salmonella* χ3306 (ZK-k) and χ3306 (ZK-kd) immunized mice orally infected with χ3306 (wild type) (*n* = 5).

## Application to *E. coli* clinical isolates

Following the promising results with *S. enterica*, the ZK-auxotrophic suicide vaccine was subsequently constructed in clinical isolates of *E. coli*. Six non-enterohemorrhagic *E. coli* strains, H19, H20, H126, N61, N67, and N77 were isolated from the milk of cows with coliform mastitis (49) and were subjected to the plasmid transformation.

First, we tested the tolerance for ZK incorporation at the UAG stop codons. All tested clinical isolates carrying plasmid A-1, which contains only the ZK incorporation system and no toxin-antitoxin genes, were viable in the presence of ZK (Fig. S1 and S4), suggesting that all tested strains are tolerant to ZK incorporation.

Next, plasmid A-3 containing the ZK incorporation system, *kid-kis,* and *ccdB-ccdA* was transformed into the parental clinical isolates. Some strains carrying plasmid A-3, H19(ZK-kd), H126(ZK-kd), N61(ZK-kd), and N67(ZK-kd) were successfully isolated although others, H20 and N77, could not be isolated even after several attempts (Fig. S5). No survivors were detected in H126(ZK-kd), N61(ZK-kd), and N67(ZK-kd) after inoculation with $5 \times 10^4$ CFU in the absence of ZK. Nonetheless, several tens of survivors were observed in H19(ZK-kd), which was excluded from further plasmid transformation.

In addition, plasmid C-1 containing *colE3-immE3* was transformed into H126(ZK-kd), N61(ZK-kd), and N67(ZK-kd). Bacteria carrying both plasmids A-3 and C-1 were successfully isolated for H126 and N61, while N67 was excluded due to natural resistance to the selection antibiotic of plasmid C-1, carbenicillin (Fig. S6).

Finally, we obtained two strains carrying the ZK-incorporation system with three inducible toxin-antitoxin pairs, N61(ZK-kdo) and H126(ZK-kdo). The escape frequencies were 0.23 and 1.09 per $10^8$ generation for N61(ZK-kdo) and H126(ZK-kdo), respectively (Fig. 2). Safety and immunogenicity were verified using N61(ZK-kdo) which satisfied the NIH standard for escape frequency. N61(ZK-kdo) was killed slightly faster after ZK-removal than BL21-AI(ZK-kdo) (Fig. 3C). The $T_{0.1}$ of N61(ZK-kdo) was estimated to be 90–105 min. Growth rates of N61(ZK-kdo) decreased to 0.65-fold that of the N61 vector control strain.

To confirm the safety of the N61(ZK-kdo) vaccine, we evaluated the safety profiles using a single-dose vaccination schedule in BALB/c mice with the N61(ZK-kdo), formalin-killed N61(ZK-kdo), or wild-type N61 at $1 \times 10^8$ CFU (Fig. 5A). In the treatment of wild-type N61 by subcutaneous injection, all mice were developed the terminal disease and died. In comparison, all mice were survived in treatment with N61(ZK-kdo), formalin-killed N61(ZK-kdo), and saline (significant $P < 0.0001$). Clinical signs in all vaccinated mice started with ruffled fur, hunched posture, decreased activity, and progressed to limb paresis or paralysis, tremor, ataxia, rigidity, dehydration, and coma occurring within 12 to 48 h of challenge, and then surviving mice recovered within 60 h (Fig. 5B; Table S2). Tissue specimens of N61(ZK-kdo)-injected mice showed only localized bacterial aggregation in the dermal tissue 24 h after inoculation. By contrast, bacterial aggregation throughout the tissue was confirmed in wild-type N61-injected mice (Fig. 5C). In addition, viable wild-type N61 bacteria were detected in the blood samples at 12 h after injection but not at 3 h, indicating bacterial proliferation after the injection (Fig. 5D). By contrast, we could not detect any viable bacteria in the samples obtained from mice injected with N61(ZK-kdo).

To measure the antibody-mediated immune responses following vaccination with N61(ZK-kdo), BALB/c mice were immunized with N61(ZK-kdo), formalin-killed N61(ZK-kdo), or saline (Fig. 5E and F). After a three-dose vaccination schedule, significant levels of total IgG against *E. coli* were present in all immunized mice on days 7 and 14 after vaccination with N61(ZK-kdo) compared to saline [significant $P < 0.05$; mean ODs in day 7, saline vs N61(ZK-kdo) = 0.15 vs 0.58; day 14, 0.19 vs 0.71], whereas no significant increase was detected in mice immunized with formalin-killed N61(ZK-kdo). On day 21 after vaccination, total IgG levels increased significantly in both N61(ZK-kdo) and formalin-killed N61(ZK-kdo) immunized mice [significant $P < 0.05$; mean ODs in day 21; saline vs formalin-killed N61(ZK-kdo) = 0.15 vs 0.52, saline vs N61(ZK-kdo) = 0.15 vs 0.78]. Notably, the live N61(ZK-kdo) was more immunogenic than the formalin-killed form [significant $P < 0.05$; mean ODs on day 7, formalin-killed N61(ZK-kdo) vs N61(ZK-kdo) = 0.21 vs 0.58; day 14, 0.31 vs 0.71; day 21, 0.52 vs 0.78].

To verify whether N61 (ZK-kdo) confers protection, we assessed efficacy using an acute systemic infection model in mice (Fig. 5G through I). A single-dose vaccination schedule with both live and formalin-killed N61 (ZK-kdo) protected mice against challenge with the wild-type N61 on day 7, whereas all control mice died within 48 h. The survival rate of vaccinated mice with formalin-killed N61 (ZK-kdo) was 60%. By contrast, all vaccinated mice with N61(ZK-kdo) were alive [significant $P = 0.0289$ for formalin-killed N61 (ZK-kdo) vs N61(ZK-kdo)]. Clinical signs in all mice started with ruffled

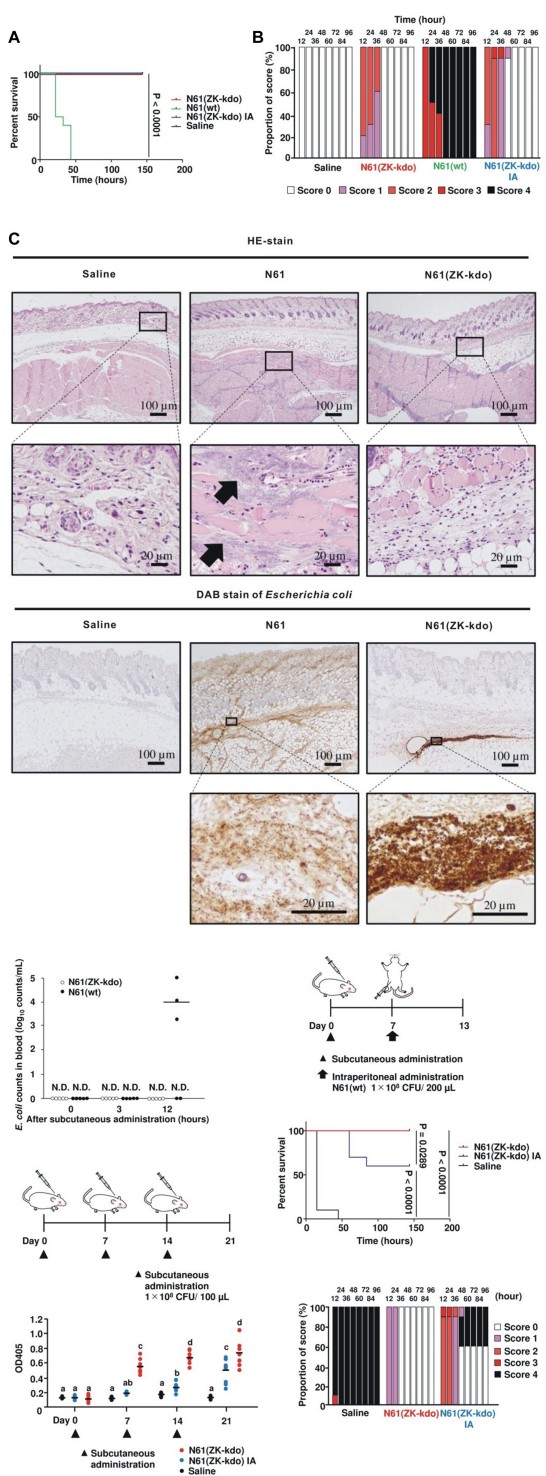

FIG 5 Application to *E. coli* clinical isolates. (A) Survival curve of mice after subcutaneous administration of N61, N61(ZK-kdo), or formalin-killed N61(ZK-kdo) strain at $1 \times 10^8$ CFU during the course of 144-h period (*n* = 10). (B) Clinical scoring results over the course of sepsis in mice that subcutaneously administrated N61, N61(ZK-kdo), or formalin-killed N61(ZK-kdo) strain. The scoring criteria are presented in Table S2. Bars show the distribution of scores for the observation parameters calculated based on scoring 1, 2, 3, and 4 at the time points after subcutaneous administration. The scores are represented by distinct colors as indicated. (C) Histological specimens of subcutaneous mouse tissues in HE stain and DAB stain of *E. coli* harvested at 24 h. Square: large scale of bottom figures. (Continued on next page)

**FIG 5** (Continued)

Arrows: bacterial aggregation. (D) Kinetics of blood clearance showing the complete elimination of N61(ZK-kdo) after subcutaneous administration in mice. White and black circles correspond to blood samples from N61(ZK-kdo) and wild-type N61 strain subcutaneously administrated mice, respectively. N.D., not detected. (E) Experimental design for evaluation of the systemic humoral responses in mice by subcutaneous administration of N61(ZK-kdo). Mice were immunized with N61(ZK-kdo) or formalin-killed N61(ZK-kdo) strain by subcutaneous administration. After three subcutaneous doses every week, anti-*E. coli*-specific IgG antibodies in serum were determined at each time point. Triangles: time point of subcutaneous administration. (F) Anti-*E. coli*-specific IgG titers in serum after subcutaneous administration of N61(ZK-kdo). Each data point represents the OD value for one sample. The bar represents the mean. Circles are represented by distinct colors as indicated. a-c: The values (average ±SD; $n = 10$) between different superscripts are significantly different ($P < 0.05$). (G) Experimental design for evaluation of the protection conferred by subcutaneous administration of N61(ZK-kdo). Mice were immunized with N61(ZK-kdo) or formalin-killed N61(ZK-kdo) strain by subcutaneous administration. After 1-week subcutaneous doses, mice were infected with the wild-type N61 strain ($1 \times 10^8$ CFU) by intraperitoneal administration. (H) Survival curves of mice administrated with N61 strain for 144 h ($n = 10$). (I) Clinical scoring results over the course of sepsis in subcutaneously administrated mice that intraperitoneally administrated N61 strain.

fur, hunched posture, decreased activity, and progressed to limb paresis or paralysis, tremor, ataxia, rigidity, dehydration, and coma occurring within 12 to 48 h after the challenge. The surviving mice recovered within 60 h.

## DISCUSSION

We have demonstrated a novel platform for quick and easy preparation of suicide vaccine candidates in *S. enterica* and *E. coli*. This platform is a transformation-based system using plasmids containing a uAA incorporation system and a toxin-antitoxin system(s) regulated by an externally supplied uAA. The vaccine candidates can be prepared within a minimum of 2 days.

The selection of the uAA incorporation system affected escape frequency. Until 2017, over 150 non-canonical amino acids, including uAAs, have been incorporated into *E. coli* proteins *in cellulo* (50). Substrate specificity and incorporation efficiency, which determine leakage expression and maximum translation level, respectively, are significantly diverse among the incorporation systems (51), suggesting the production of antitoxin, which is regulated by the incorporation of uAAs, may be affected. In the IY incorporation system, a modified system with better switchability using a positive feedback loop reduced the escape frequency to 0.1-fold. The modified system suppresses leakage expression, indicating that lowered antitoxin expression level in the absence of IY enhances killing by the cognate toxin. The exchange of the modified IY incorporation system with the ZK incorporation system further reduced the escape frequency to 0.1-fold. In addition, both the initial proliferation period and duration of viable bacteria after the uAA removal were also reduced. Since the switchability of the modified IY and ZK incorporation systems is at a similar level, a difference in biological or chemical properties between IY and ZK, or a lower efficiency of the ZK incorporation is a possible cause, which could affect the stability, activity, or expression level of the antitoxins.

Incorporation of uAA sometimes severely damages host bacteria (52). However, all the tested strains, including the *E. coli* laboratory strain BL21-AI, *E. coli* clinical isolates, and *S. enterica* χ3306, were tolerant to the incorporation of ZK at the UAG stop codon. This observation suggests that the disorder caused by the uAA incorporation may not be a frequently occurring barrier, whereas more extensive research would be needed.

Selection and combination of the toxin-antitoxin systems was crucial in determining the vaccine characteristics. We showed that, in addition to previously used ColE3-ImmE3, the toxicity of Kid-Kis and CcdB-CcdA can be controlled by ZK in *E. coli* BL21-AI, suggesting that the repertoire of available toxin-antitoxins is scalable. However, plasmid

A3 containing both *kid-kis* and *ccdB-ccdA* failed to be transformed into the *E. coli* clinical isolates H20 and N77, suggesting that the toxicity was not completely suppressed in these strains even in the presence of ZK. Similarly, *S. enterica* χ3306 was unable to maintain plasmid C-1 containing *colE3-immE3*. Conversely, unusually many survivors were detected in H19 carrying plasmid A3, presumably due to lower toxin or predominant antitoxin activity. These observations suggest that the toxin-antitoxin systems working well in BL21-AI are not always available in other *E. coli* and *S. enterica* strains. This problem caused the failure to achieve sufficiently low escape frequencies in some *E. coli* clinical isolates and *S. enterica* χ3306. The resulting strains, in the present form, are not suitable for use in an open environment. The gene organization including regulatory regions was maintained as wild type in the toxin-antitoxin genes because the expression intensity of these genes is optimally balanced by evolutionary selection for stable maintenance through generations (13, 53). The optimal gene organization in one strain, nonetheless, may not necessarily be the case in other strains. Moreover, the antitoxin protein production could be attenuated due to competition between ZK-tRNA$_{CUA}$ and peptide release factor 1 at the UAG stop codons (54, 55), thereby disturbing the optimal balance between toxins and antitoxins. A possible solution is to use a wide variety of toxin-antitoxin genes and a library of mutants of their regulatory regions. The promising vaccine candidates with desired characteristics may be obtained after screening the library transformants.

All strains carrying only a single toxin-antitoxin were unable to reach the escape frequencies below $1 \times 10^{-8}$. Some toxin-antitoxins, nonetheless, were available in this study, enabling for construction of multilayered systems. Finally, we achieved to generate strains with low escape frequencies below $1 \times 10^{-8}$ using a combination of two or three toxin-antitoxins, suggesting that the multilayered system is an effective strategy also in our method, as suggested in previous studies on biological containment systems (46, 47).

The duration of bacterial viability after ZK removal was affected by the selection of toxin-antitoxins. In BL21-AI, the strain carrying both *kid-kis* and *ccdB-ccdA* survived for relatively long time, but the additional transformation of *colE3-immE3* dramatically reduced the duration. Interestingly, the reduced duration was almost identical to that of the *colE3-immE3* single transformant. In this case, the duration should be determined by the toxicity of the specific toxin-antitoxin rather than the sum of the effects of multiple toxin-antitoxins since a single toxin is enough to kill the host. This observation is of great significance in selecting the appropriate toxin-antitoxin combination to achieve the intended duration. χ3306(ZK-kd) continued to proliferate and survived remarkably longer after ZK removal than BL21-AI(ZK-kd), suggesting that host-dependent differential responses are another important factor.

All the vaccine candidate strains showed impaired growth rates compared to the vector control. This secondary attenuation should be considered as a factor affecting vaccine efficacy.

Safety is one of the most important characteristics since we intend to convert pathogens directly into vaccine candidates. Mice survived after inoculation with *S. enterica* χ3306(ZK-k) and χ3306(ZK-kd) at $1 \times 10^{5}$ CFU although the LD$_{50}$ value of wild-type χ3306 is $<1 \times 10$ CFU. Wild-type *E. coli* N61 killed mice at $1 \times 10^{8}$ CFU and formed bacterial colonies near the injection site. In addition, viable bacteria were detected in blood 12 h after injection indicating bacterial proliferation *in vivo*. By contrast, mice survived after the injection of N61(ZK-kdo) at the same dose. No colony formation or proliferation was observed *in vivo*. These results suggest that the ZK-auxotrophic suicidal vaccine candidates are much less virulent than the parental wild-type pathogens in both *S. enterica* and *E. coli* clinical isolates.

Another critical characteristic is immunogenicity. Specific IgG induction and protection against the parental wild-type pathogens were confirmed in both *S. enterica* and *E. coli*. Two important observations were noted. First, protection against oral challenge, as well as intraperitoneal challenge, was observed in *S. enterica*. Second, the *E. coli* suicide live vaccine induced stronger IgG production and showed more

potent protection than the formalin-inactivated vaccine. These findings suggest that the strong immunogenicity characteristic of live vaccines is achievable in the uAA-regulated conditional suicide vaccines.

We selected two bacterial pathogens, *S. enterica* and *E. coli*, as models for vaccine construction. *S. enterica* encompasses *S. typhi* and *S. paratyphi,* which, respectively, cause typhi and paratyphi fever (56). Non-typhoidal salmonella is also a major public health problem in Africa, killing ~49,600 people each year (57). *S. enterica* is pathogenic in animals as well as humans. Domestic salmonellosis has been reported in cattle, swine, and chickens. Most cases of salmonellosis are caused by the consumption of contaminated eggs, chicken, pork, beef, and dairy products, although other animals such as rats and companion animals have also been identified as sources of infection in humans. *S. enterica* has the potential to be drug resistant, and vaccination is an effective measure for controlling infections. However, no universal vaccine is currently available for humans and animals. Animal studies have shown that attenuated live vaccines are more effective but they may pose a danger to immunodeficiency patients and children. χ3306(ZK-k) and χ3306(ZK-kd)-immunized mice showed protection against infection both peritoneally and orally. Notably, the vaccination exerted a better protective effect against oral challenge than intraperitoneal challenge (Fig. 4D and G). *S*. Typhimurium commonly transmits *via* the oral route (58), suggesting that the uAA-auxotrophic suicide vaccine system has the potential to be put in practical use. Furthermore, these vaccine strains were dramatically less virulent than wild-type χ3306. These results suggest that the uAA-auxotrophic suicide vaccine represents a promising approach for developing safe and highly immunogenic *Salmonella* vaccines.

Pathogenic *E. coli,* which has acquired specific combinations of virulence factors, causes various diseases in healthy individuals, including enteric/diarrheal disease, urinary tract infections, and sepsis/meningitis (32). *E. coli* is also an important pathogen in animals. *E. coli* mastitis is one of the most common diseases in the dairy industry and can cause severe acute inflammation with fever and decreased milk production, and lead to serious economic losses (59–62). The virulence factors of the pathogenic *E. coli,* which include LPS among others, have been documented (63). During *E. coli* infection and multiplication in the mammary gland, the release of LPS activates the host's immune system (64). Although immune cells can invade and destroy LPS, a large amount of LPS is released and causes systemic symptoms in the host (65). Currently, the vaccine used to prevent *E. coli* bovine mastitis is the J5 vaccine which is an inactivated *E. coli* vaccine with an incomplete O-antigen (65). A previous study showed that the J5 immune serum was not an improvement on the already high efficiency of naturally acquired antibodies against *E. coli* (66). Moreover, the safety and effectiveness of the J5 vaccine in dairy cows have not been clearly reported (67). The use of killed *E. coli* vaccines including autovaccines has not produced the desired effects (68, 69). The attenuation procedure should be performed in a manner that prevents the vaccine from causing a persistent carrier state (70). This can be achieved by N61(ZK-kdo) strain's ability to colonize and proliferate in the host for a limited period being eliminated without causing disease. This was observed for the N61(ZK-kdo) when injected intraperitoneally into mice, as it was completely cleared from the blood. N61(ZK-kdo) demonstrated superior vaccine efficacy compared to formalin-inactivated N61(ZK-kdo), suggesting that this innovative vaccine is a possible strategy to protect against various human and animal colibacilloses including bovine coliform mastitis.

Translational research is needed to mature the system to facilitate both safety and immunogenicity for a wide range of strains for clinical use. Exploration of available uAA incorporation systems and toxin-antitoxin systems, and selection of optimal vaccine strains through library screening would be promising strategies. Various routes of vaccine administration, including intranasal and oral administration, should also be considered. In addition, limited data on the safety of new technologies such as our study may also raise hesitancy or refusal to their practical use. Future work is desired to include

efforts to collect data from non-clinical and clinical trials that will demonstrate effective and safe new vaccines.

In conclusion, we demonstrated the basic concept of a novel strategy for quick and easy conversion of bacterial pathogens into the uAA-auxotrophic suicide vaccine candidates using conditionally toxic plasmids. Users can control the critical vaccine characteristics such as the escape frequency and the duration of bacterial viability after the uAA removal by selecting of the uAA incorporation system and the toxin-antitoxins. This platform should enable the rapid preparation of vaccine candidate strains against various infectious diseases caused by *S. enterica* and *E. coli* for which highly effective vaccines have not yet been developed. The concept of the uAA-auxotrophic suicide vaccine may be scalable for other microbial pathogens after transplanting the uAA incorporation systems and developing the applicable toxin-antitoxin systems.

## MATERIALS AND METHODS

### Strains, growth conditions, and transformation

*E. coli* BL21-AI was commercially purchased. Six non-enterohemorrhagic *E. coli* strains, H19, H20, H126, N61, N67, and N77, were isolated from the milk of cows with coliform mastitis in Hokkaido, Japan. The complete genomic sequence of the N61 strain has been deposited in DDBJ/EMBL/GeneBank under accession number AP028880. *E. coli* XL1-Blue and *E. coli* DH5α were also used for plasmid construction. All *E. coli* strains were grown in Luria-Bertani (LB) medium or dilutions at 37°C and 200 rpm for all experiments. *Salmonella enterica* serovar Typhimurium (*S.* Typhimurium) χ3306 was grown in LB medium at 37°C and standing culture for all experiments. Agar (final 2%) was added for the preparation of the solid medium. Carbenicillin (100 µg/mL) and chloramphenicol (50 µg/mL) were added as appropriate. The transformation was performed by electroporation using a Gene Pulser II electroporator (BIO-RAD).

### Plasmid construction

Maps for all plasmids are shown in Fig. S1. Plasmids A-1 and B-1 are identical to pTYR MjIYRS2-1(D286) MJR1 ×3 and pTK2-1 ZLysRS1, respectively, kindly provided from Kensaku Sakamoto and Shigeyuki Yokoyama (RIKEN) (41, 43). Plasmid C-1 is pSH350 distributed by Haruhiko Masaki (38). Plasmids A-2, A-3, and C-2 were constructed using gene synthesis outsourced to GenScript (Tokyo, Japan). Plasmid B-2 is 2a-IYRS(M6V) described in our previous paper (42).

### Escape frequency

Escape frequency was estimated by fluctuation assay as previously described (13, 71), with some modifications. Briefly, 10 parallel cultures were prepared using twofold diluted LB medium containing 3 mM ZK for *E. coli* strains and 1 mM ZK for *S.* Typhimurium strains and incubated for approximately 16 h (final $OD_{590}$ = 0.03–0.1). The bacteria were collected by centrifugation, resuspended in 1 mL of ZK-free LB medium, and centrifuged again for washing. The wash was repeated three times to remove ZK completely. The bacteria were resuspended and inoculated onto a ZK-free solid medium. After overnight culture, escapers were detected as colonies. To estimate a total number of inoculated bacteria, the bacterial suspension was also plated onto a solid medium containing 3 mM ZK for *E. coli* strains and 1 mM ZK for *S.* Typhimurium strains. The mutation rates were calculated by the web tool FALCOR using the Ma-Sandri-Sarkar Maximum Likelihood Estimator method (https://lianglab.brocku.ca/FALCOR/) (72).

### Duration of bacterial viability after ZK removal

Bacteria were cultured in LB medium diluted to 1/4 concentration containing 3 mM ZK for *E. coli* strains and 1 mM ZK for *S.* Typhimurium strains and appropriate selection marker antibiotics for 16 h. To obtain the bacteria at the logarithmic growth phase, the

culture medium was exchanged with fresh ZK-containing LB medium. The bacteria were additionally cultured for 1.5 h. After washing twice with fresh LB medium, the bacterial suspension was diluted $10^5$-fold in 30 mL of ZK-free LB medium and incubated. An aliquot (250 µL) of the bacterial culture was directly inoculated onto a ZK-containing LB agar plate. After overnight culture, appearing colonies were counted.

## Growth rate

The bacterial culture was prepared as described above and diluted in 50 mL of LB medium containing 3 mM ZK for *E. coli* strains and 1 mM ZK for *S.* Typhimurium strains to calculate $OD_{590}$ = 0.01–0.03. The diluted culture was incubated for 1–2 h until $OD_{590}$ reached to 0.6–1.2. An aliquot (1 mL) of the bacterial culture was withdrawn at 15-min interval to measure bacterial density as $OD_{590}$. A single fitted curve was generated for each strain using Origin7.

## Animals

Six-week-old female BALB/c mice were purchased from SLC Japan, Inc. (Hamamatsu, Japan) and Hokudo Co. (Sapporo, Japan). Animals were maintained and used in accordance with the Guidelines for the Care and Use of Laboratory Animals of the National Institute of Animal Health. All animal procedures were carried out in strict accordance with local guidelines and with ethical approval from the National Institute of Animal Health (authorization numbers: 19-035 and 20-081). Blood samples were collected from the submandibular vein of anesthetized mice, and sera were separated from the blood cells by centrifugation (1,500 × $g$, 15 min) and stored at −80 °C until subsequent analysis.

### *Salmonella* infection experiments

Six-week-old mice were immunized by intraperitoneal and intravenous administration of χ3306 (ZK-k) or χ3306 (ZK-kd), all challenges were made using the intraperitoneal administration route and oral challenge with *S.* Typhimurium χ3306. Intraperitoneal infections were given at a dose of approximately 10 CFU/200 µL ($LD_{50}$ in BALB/c: <10 CFU) (73, 74). Oral infections were administered at approximately $10^6$ CFU/20 µL ($LD_{50}$ in BALB/c: <$10^4$ CFU) (75).

### *E. coil* infection experiments

To prepare the formalin-inactivated *E. coli* N61 (ZK-kdo), the washed bacteria were resuspended in phosphate-buffered saline (PBS) containing 0.5% formaldehyde, incubated with shaking (50 rpm) at room temperature for inactivation, and then washed three times with PBS. After 1 h following inactivation, the cells were checked by plating on LB agar to determine the absence of colonies after overnight incubation. The inactivation step was then performed overnight to ensure complete inactivation.

All immunizations were made using the subcutaneous administration, all challenges were performed by the intraperitoneal administration route of administration. N61 and N61 (ZK-kdo) strains were cultured in LB with 10 mM ZK at 37 °C and 180 rpm, harvested (8,000 × $g$, 15 min), washed twice with LB, and finally adjusted with saline. For intraperitoneal and subcutaneous administration of N61 and N61 (ZX-kdo) strains, we used a total volume of 200 and 100 µL, respectively. During the subcutaneous administration or infection with *E. coli,* mice were monitored twice daily and a final disease score was given to each mouse according to the clinical signs observed as summarized in Table S2 (76).

## Histological analysis

Tissue samples collected from the subcutaneous administration site were fixed in 10% phosphate-buffered formalin and embedded in paraffin for histochemical analysis. Paraffin sections (4 µm thick) were mounted on slide glasses, de-waxed in xylene,

rehydrated through a series of graded ethanol solutions, and transferred to PBS (pH 7.4). Tissue sections were stained with hematoxylin and eosin (HE). Immunostaining was performed using the peroxidase diaminobenzidine (DAB) method, by means of a Simple Stain Max kit (Nichirei, Tokyo, Japan) with primary rabbit polyclonal antibody to *E. coli* (1:5,000; Dako A/S, Glostrup, Denmark). The sections were observed using a light microscope (Olympus, DP27, Tokyo, Japan).

## Enzyme-linked immunosorbent assay for the detection of specific IgG antibodies

For enzyme-linked immunosorbent assay (ELISA), a 96-well plate (Nunc; Roskilde, Denmark) was coated with 50 µL of 10 µg/mL *S*. Typhimurium χ3306 LPS extracted using the LPS extraction kit (iNtRON Biotechnology, Inc.) and diluted in 50 mM carbonate-bicarbonate buffer, pH 9.6. All incubations were carried out at 37°C for 60 min, and every incubation step was followed by four washes with ELISA wash buffer (0.9% NaCl supplemented with 0.1% Tween-20). After coating and washing, the plates were incubated with serum samples diluted 1:100 in PBST (PBS with 0.1% Tween-20). Bound antibodies were detected using goat anti-mouse IgG conjugated to horseradish peroxidase (HRP; Southern Biotech, Birmingham, AL, USA) diluted 1:2,000 in PBST and developed using 3,3′,5,5′-tetramethylbenzidine substrate (Thermo Scientific Pierce, Rockland, IL, USA). Absorbance was measured at 450 nm using an ELISA plate reader.

To detect specific IgG antibodies against *E. coli* in serum, a microtiter plate was directly coated with formalin-killed wild-type N61 as a capture antigen. Formalin-killed N61 in PBS was dried in an oven at $5 \times 10^6$ cells/well in 96-well ELISA plates (C96 Maxisorp cert, Nunc-Immuno Plate, Thermo Fisher Scientific) overnight at 37°C. After incubation, wells were washed with Tris-buffered saline-Tween-20 (TBST) and then incubated with 100 µL of milk (diluted 1:100 in PBS) for 90 min at room temperature. After five TBST washes, wells were incubated with horseradish peroxidase-conjugated sheep anti-mouse IgG antibody (diluted 1:30,000, Bethyl Laboratories, Inc., Montgomery, TX, USA) for 2 h at room temperature. The freshly prepared substrate was added, and OD was measured at 405 nm using the 3,3′,5,5′-tetramethylbenzidine microwell peroxidase substrate system (KPL, Gaithersburg, MD, USA). All samples were analyzed in duplicate and mean values were calculated. Anti-*E. coli*-IgG antibodies were calculated by subtracting the OD values for the buffer controls ($OD_{405}$ = ~0.10), which were included in duplicate in all ELISAs, from the specific sample OD values. To standardize and compare results between plates, positive control milk samples (from murine serum IP by *E. coli*, $OD_{405}$ = ~1.0 for specific IgG) were included in duplicate in all ELISAs. OD values were normalized against those of positive controls.

## Statistical analysis

The one-way analysis of variance (ANOVA) was used for multiple comparisons, followed by Bonferroni's post hoc test using SPSS software (IBM SPSS Statistics version 25, Tokyo, Japan). Survival data were compared using the log-rank test and Wilcoxon test using Prism 6.0 (GraphPad Prism, GraphPad Software, Inc.). Differences with *P* values less than 0.05 were considered significant.

## ACKNOWLEDGMENTS

We thank Kensaku Sakamoto and Shigeyuki Yokoyama (RIKEN) for plasmids A-1 and B-1 and Haruhiko Masaki (Tokyo University) for plasmid C-1.

This work was supported by JSPS grant number 20H03158.

Y.K. designed the project; Y.K. designed microbial experiments; M.E. and M.N. designed *S. enterica* vaccine experiments; Y.N. and T.H. designed *E. coli* vaccine experiments; Y.N., M.N., Y.K., Y.O., S.D.A., Y.T., T.I., O.M., A.S., M.O., T.H., and M.E. performed the research; Y.N., M.N., Y.K., T.H., and M.E. analyzed the data; Y.K., Y.N., M.N., and M.E. wrote the paper. All authors reviewed drafts of the paper.

## AUTHOR AFFILIATIONS

[1]National Institute of Animal Health, National Agriculture and Food Research Organization (NARO), Sapporo, Hokkaido, Japan

[2]National Institute of Animal Health, National Agriculture and Food Research Organization (NARO), Tsukuba, Ibaraki, Japan

[3]Institute of Agrobiological Sciences, National Agriculture and Food Research Organization (NARO), Tsukuba, Ibaraki, Japan

## PRESENT ADDRESS

Tomohito Hayashi, Nihon Zenyaku Kogyo, Co., Ltd., Koriyama, Fukushima, Japan

## AUTHOR ORCIDs

Yuya Nagasawa http://orcid.org/0000-0003-1392-6454
Momoko Nakayama http://orcid.org/0000-0002-9284-8425
Yusuke Kato http://orcid.org/0000-0003-2424-4571
Yohsuke Ogawa http://orcid.org/0000-0003-4107-9694
Tomohito Hayashi http://orcid.org/0000-0003-1631-1017
Masahiro Eguchi http://orcid.org/0009-0008-5090-7221

## FUNDING

| Funder | Grant(s) | Author(s) |
|---|---|---|
| MEXT | Japan Society for the Promotion of Science (JSPS) | 20H03158 | Yusuke Kato |

## AUTHOR CONTRIBUTIONS

Yuya Nagasawa, Investigation, Writing – original draft | Momoko Nakayama, Investigation, Writing – original draft | Yusuke Kato, Conceptualization, Funding acquisition, Investigation, Project administration, Resources, Supervision, Writing – original draft, Writing – review and editing | Yohsuke Ogawa, Investigation | Swarmistha Devi Aribam, Investigation | Yusaku Tsugami, Investigation | Taketoshi Iwata, Investigation | Osamu Mikami, Investigation | Aoi Sugiyama, Investigation | Megumi Onishi, Investigation | Tomohito Hayashi, Investigation, Supervision | Masahiro Eguchi, Investigation, Supervision, Writing – original draft

## ADDITIONAL FILES

The following material is available online.

### Supplemental Material

**Supplemental material (Spectrum03557-23-S0001.pdf).** Figures S1 to S6; Tables S1 and S2.

### Open Peer Review

**PEER REVIEW HISTORY (review-history.pdf).** An accounting of the reviewer comments and feedback.

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
