## [Reviewer comments · Microbiology Spectrum]

Microbiology Spectrum

A novel vaccine strategy using quick and easy conversion of bacterial pathogens to unnatural amino acid-auxotrophic suicide derivatives

Yuya Nagasawa, Momoko Nakayama, Yusuke Kato, Yohsuke Ogawa, Swarmistha Aribam, Yusaku Tsugami, Taketoshi Iwata, Osamu Mikami, Aoi Sugiyama, Megumi Onishi, Tomohito Hayashi, and Masahiro Eguchi

Corresponding Author(s): Yusuke Kato, Nogyo Shokuhin Sangyo Gijutsu Sogo Kenkyu Kiko

Review Timeline:

Submission Date:	October 4, 2023
Editorial Decision:	December 4, 2023
Revision Received:	January 23, 2024
Accepted:	January 24, 2024

Editor: Artem Rogovskyy

Reviewer(s): Disclosure of reviewer identity is with reference to reviewer comments included in decision letter(s). The following individuals involved in review of your submission have agreed to reveal their identity: Fei Kean Loh (Reviewer #1); Jeffery Tharp (Reviewer #3)

Transaction Report:

DOI: <https://doi.org/10.1128/spectrum.03557-23>

Re: Spectrum03557-23 (A novel vaccine strategy using quick and easy conversion of bacterial pathogens to unnatural amino acid-auxotrophic suicide derivatives)

Dear Dr. Yusuke Kato:

Thank you for the privilege of reviewing your work. Below you will find my comments, instructions from the Spectrum editorial office, and the reviewer comments.

Revision Guidelines

Sincerely,
Artem Rogovskyy
Editor
Microbiology Spectrum

Reviewer #1 (Comments for the Author):

General:

Authors demonstrated great efforts in designing suicide live vaccine candidates, which adapt both uAA incorporation system and toxin-antitoxin system. The employment of multi-layer toxin-antitoxin systems is an interesting approach results in low escape frequency. The novelty of this approach is unknown. The major concern is regarding the inconsistent administration route,

dosage concentration and frequency of the bacteria and vaccine strains used throughout the study. The sample size is small. Despite the live vaccines tested here confer partial protection and IgG production, it is not conclusive to be translated for clinical use. More works can be done to profile the immune responses induced by the vaccine candidates, such as measurement of cytokines, phenotyping of immune cells using flow cytometry, or RNA sequencing etc. The results may serve as preliminary findings, which authors should propose future plan on improving these strains. Authors should be more moderate in using description words, avoid using "most", "extremely", "excellent", etc.

Importance:

- Suggest to emphasize on the unique advantages of using the proposed conditional killing system in live bacterial vaccine.

Introduction:

- Suggest to briefly compare the approach in this study with other conditional suicide systems.

Result:

- Did authors confirm the expression of the integrated plasmids and proteins encoded?
- How many independent experiments were conducted to generate the escape frequency?
- "To further reduce the escape frequency, we attempted to transform plasmid C-1 into χ 3306 (ZK-k) and χ 3306 (ZK-kd), but no transformants were obtained."

Any further improvement to solve this problem? Would these strains still be practical to use if their escape frequencies do not meet NIH requirement?

- Authors should justify the use of BALB/c mouse strain in this study. Why the infection was done via intravenous administration?
 - "After fifth-dose vaccination, double-dose administration at 1×10^4 CFU/mouse following triple-dose administration at 1×10^3 CFU/mouse for χ 3306 (ZK-k) or χ 3306 (ZK-kd), the immunized mice were challenged with intraperitoneal injection of the wild-type χ 3306 at 1×10 CFU/mouse."
- Why the CFU dosage is different for each vaccination?
- " χ 3306 (ZK-kd) induced significantly more IgG than χ 3306 (ZK-k)."
- Authors should be more precise in describing the result, by including the value, significance, fold-change. This comment applies to all results.

- "Mice were intraperitoneally vaccinated with a double-dose administration at 1×10^5 CFU/mouse of χ 3306 (ZK-k) or χ 3306 (ZK-kd)."

Justify the dosage use.

- Fig 4, panel arrangement can be improved, either from left to right or top to bottom.
- Fig 4 A, why the survival day was short (max 6 days)?
- Fig 4 C and F, the standard unit for IgG titers should be IU/ml? Significant values and statistical tests for this experiment are not mentioned.
- Fig 5 C, histology stains are not clear. Suggest to show overview tissue stain, and high resolution of area that show bacterial aggregation.

Discussion:

- Instead of using "quick and easy preparation", suggest to use more precise, for example, the vaccine candidate can be prepared in how many days etc.
- Suggest to include literature or evidence from previous studies to support the paragraph mentioned "The selection of the uAA incorporation system affected escape frequency."
- Authors should discuss why the vaccine candidates only confer partial protection?
- The limitation of study was not discussed.

Materials and Methods:

- Any reason for using young age (3-week-old) animal?

Reviewer #3 (Comments for the Author):

This paper describes a novel approach for generating live-attenuated vaccines for bacterial pathogens using genetic engineering. Live attenuated vaccines are desirable because they are often more immunogenic and provide greater protection than other methods of vaccination; however, major problems can arise if the pathogen used as a vaccine is not effectively attenuated. In this paper, the authors used toxin-antitoxin genes to generate *E. coli* (laboratory and pathogenic strains) and *Salmonella enterica*, that can only survive in media supplemented with an unnatural amino acid (uAA). These engineered bacteria can be grown in media containing a uAA, which facilitates stop codon readthrough, and expression of an antitoxin gene. The cultured cells can then be used to inoculate an animal as a vaccine. Because the uAA does not exist within the animal, the antitoxin gene is not expressed, and the microbes die shortly after injection. Importantly, the bacteria are still able to illicit an immune response inducing the animal to generate antibodies to the pathogen. Building on their previous work, herein, the authors refine this method of generating bacteria that are auxotrophic for a uAA and successfully demonstrate the use of this technology for immunizing laboratory mice against pathogenic strains of bacteria.

This paper is well written and technically sound. The data are sufficient to support the conclusions made. I would rate the novelty of the work as moderate-uAAs have been used to generate live attenuated vaccines previously, as have toxin-antitoxin genes. Thus, this work is somewhat incremental. Moreover, it is limited to bacterial strains for which multiple expression plasmids are readily available. However, given the quality of the work and the pressing needs for rapid and effective methods for generating vaccines, I believe the work will be of interest to a diverse audience. I find no major issues that the authors must address at this time.

Reviewer Comment:

General:

Authors demonstrated great efforts in designing suicide live vaccine candidates, which adapt both uAA incorporation system and toxin-antitoxin system. The employment of multi-layer toxin-antitoxin systems is an interesting approach results in low escape frequency. The novelty of this approach is unknown. The major concern is regarding the inconsistent administration route, dosage concentration and frequency of the bacteria and vaccine strains used throughout the study. The sample size is small. Despite the live vaccines tested here confer partial protection and IgG production, it is not conclusive to be translated for clinical use. More works can be done to profile the immune responses induced by the vaccine candidates, such as measurement of cytokines, phenotyping of immune cells using flow cytometry, or RNA sequencing etc. The results may serve as preliminary findings, which authors should propose future plan on improving these strains. Authors should be more moderate in using description words, avoid using “most”, “extremely”, “excellent”, etc.

Importance:

- Suggest to emphasize on the unique advantages of using the proposed conditional killing system in live bacterial vaccine.

Introduction:

- Suggest to briefly compare the approach in this study with other conditional suicide systems.

Result:

- Did authors confirm the expression of the integrated plasmids and proteins encoded?
- How many independent experiments were conducted to generate the escape frequency?
- “To further reduce the escape frequency, we attempted to transform plasmid C-1 into χ 3306 (ZK-k) and χ 3306 (ZK-kd), but no transformants were obtained.”
Any further improvement to solve this problem? Would these strains still be practical to use if their escape frequencies do not meet NIH requirement?
- Authors should justify the use of BALB/c mouse strain in this study. Why the infection was done via intravenous administration?
- “After fifth-dose vaccination, double-dose administration at 1×10^4 CFU/mouse following triple-dose administration at 1×10^3 CFU/mouse for χ 3306 (ZK-k) or χ 3306 (ZK-kd), the immunized mice were challenged with intraperitoneal injection of the wild-type χ 3306 at 1×10 CFU/mouse.”
Why the CFU dosage is different for each vaccination?
- “ χ 3306 (ZK-kd) induced significantly more IgG than χ 3306 (ZK-k).”

Authors should be more precise in describing the result, by including the value, significance, fold-change. This comment applies to all results.

- “Mice were intraperitoneally vaccinated with a double-dose administration at 1×10^5 CFU/mouse of $\chi 3306$ (ZK-k) or $\chi 3306$ (ZK-kd).”
Justify the dosage use.
- Fig 4, panel arrangement can be improved, either from left to right or top to bottom.
- Fig 4 A, why the survival day was short (max 6 days)?
- Fig 4 C and F, the standard unit for IgG titers should be IU/ml? Significant values and statistical tests for this experiment are not mentioned.
- Fig 5 C, histology stains are not clear. Suggest to show overview tissue stain, and high resolution of area that show bacterial aggregation.

Discussion:

- Instead of using “quick and easy preparation”, suggest to use more precise, for example, the vaccine candidate can be prepared in how many days etc.
- Suggest to include literature or evidence from previous studies to support the paragraph mentioned “The selection of the uAA incorporation system affected escape frequency.”
- Authors should discuss why the vaccine candidates only confer partial protection?
- The limitation of study was not discussed.

Materials and Methods:

- Any reason for using young age (3-week-old) animal?

This paper describes a novel approach for generating live-attenuated vaccines for bacterial pathogens using genetic engineering. Live attenuated vaccines are desirable because they are often more immunogenic and provide greater protection than other methods of vaccination; however, major problems can arise if the pathogen used as a vaccine is not effectively attenuated. In this paper, the authors used toxin-antitoxin genes to generate *E. coli* (laboratory and pathogenic strains) and *Salmonella enterica*, that can only survive in media supplemented with an unnatural amino acid (uAA). These engineered bacteria can be grown in media containing a uAA, which facilitates stop codon readthrough, and expression of an antitoxin gene. The cultured cells can then be used to inoculate an animal as a vaccine. Because the uAA does not exist within the animal, the antitoxin gene is not expressed, and the microbes die shortly after injection. Importantly, the bacteria are still able to illicit an immune response inducing the animal to generate antibodies to the pathogen. Building on their previous work, herein, the authors refine this method of generating bacteria that are auxotrophic for a uAA and successfully demonstrate the use of this technology for immunizing laboratory mice against pathogenic strains of bacteria.

This paper is well written and technically sound. The data are sufficient to support the conclusions made. I would rate the novelty of the work as moderate—uAAs have been used to generate live attenuated vaccines previously, as have toxin-antitoxin genes. Thus, this work is somewhat incremental. Moreover, it is limited to bacterial strains for which multiple expression plasmids are readily available. However, given the quality of the work and the pressing needs for rapid and effective methods for generating vaccines, I believe the work will be of interest to a diverse audience. I find no major issues that the authors must address at this time.

Response to Reviewer 1:

First, we thank the reviewers for their constructive comments. Our manuscript has been revised in response to the following individual remarks.

General:

Authors demonstrated great efforts in designing suicide live vaccine candidates, which adapt both uAA incorporation system and toxin-antitoxin system. The employment of multi-layer toxin-antitoxin systems is an interesting approach results in low escape frequency. The novelty of this approach is unknown. The major concern is regarding the inconsistent administration route, dosage concentration and frequency of the bacteria and vaccine strains used throughout the study. The sample size is small. Despite the live vaccines tested here confer partial protection and IgG production, it is not conclusive to be translated for clinical use. More works can be done to profile the immune responses induced by the vaccine candidates, such as measurement of cytokines, phenotyping of immune cells using flow cytometry, or RNA sequencing etc. The results may serve as preliminary findings, which authors should propose future plan on improving these strains. Authors should be more moderate in using description words, avoid using “most”, “extremely”, “excellent”, etc.

This is our first paper on the uAA-auxotrophic suicide vaccine. Thus, the main goal is to demonstrate the basic concepts. Detailed analysis of the immune responses elicited by individual vaccine candidates and attempts to improve the system are considered future work. As suggested by the reviewer, we have added a note about future research plans (page 18, line 6-14). We have also reconsidered the words used.

Importance:

Suggest to emphasize on the unique advantages of using the proposed conditional killing system in live bacterial vaccine.

Following the reviewer's suggestion, we have made an additional note (page 3, line 35-page 4, line 2).

Introduction:

Suggest to briefly compare the approach in this study with other conditional suicide systems.

Following the reviewer's suggestion, we have made an additional note (page 6, line 18-21).

Result:

Did authors confirm the expression of the integrated plasmids and proteins encoded?

The expression of the integrated plasmids and proteins in the vaccine candidates was supported by the phenotype, uAA-auxotrophy (Fig. S2, S3 and S4). There is no proper tool such as antibodies against the proteins, so that, unfortunately, protein expression analysis could not be performed.

□ How many independent experiments were conducted to generate the escape frequency?

The escape frequency was evaluated by a fluctuation assay. This assay used 10 biologically independent parallel cultures. The escape frequency was statistically calculated from the distribution of results obtained from the 10 samples.

□ “To further reduce the escape frequency, we attempted to transform plasmid C-1 into χ 3306 (ZK-k) and χ 3306 (ZK-kd), but no transformants were obtained.”

Any further improvement to solve this problem? Would these strains still be practical to use if their escape frequencies do not meet NIH requirement?

Vaccine candidates, that do not achieve sufficiently low escape frequencies, are not expected to be clinically applicable as is. To clarify this point, we have made an additional note (page 15, line 7-10). A possible solution is described at the end of paragraph.

□ Authors should justify the use of BALB/c mouse strain in this study. Why the infection was done via intravenous administration?

BALB/c mice are commonly used as an experimental model for E. coli and Salmonella infection. This strain is known to be more susceptible to Salmonella infection than other inbred mice.

Intravenous administration is one method of administering Salmonella to mice. It is more susceptible than oral infection, and intravenous administration was used to test the difference in susceptibility between the wild strain and the ZK-kd or ZK-d strains. The citations for oral infection, peritoneal infection, and intravenous administration were added to the Methods section.

□ “After fifth-dose vaccination, double-dose administration at 1×10^4 CFU/mouse following triple-dose administration at 1×10^3 CFU/mouse for χ 3306 (ZK-k) or χ 3306 (ZK-kd), the immunized mice were challenged with intraperitoneal injection of the wild-type χ 3306 at 1×10 CFU/mouse.”

Why the CFU dosage is different for each vaccination?

The first immunization was set to 1×10^3 CFU/mouse to avoid the onset of septic shock due to LPS.

□ “ χ 3306 (ZK-kd) induced significantly more IgG than χ 3306 (ZK-k).”

Authors should be more precise in describing the result, by including the value, significance, fold-change. This comment applies to all results.

The following text has been modified to the manuscript.

Page 10, line 17-28:

After fourth-dose vaccination, single-dose administration at 1×10^3 CFU/mouse following triple-dose administration at 1×10^4 CFU/mouse for χ 3306 (ZK-k) or χ 3306 (ZK-kd), the immunized mice were challenged with intraperitoneal injection of the wild-type χ 3306 at 1×10 CFU/mouse (Figure 4B). We evaluated the induction of anti-*S. enterica* lipopolysaccharide (LPS) IgG after the intraperitoneal administration with χ 3306 (ZK-k) and χ 3306 (ZK-kd) (Figure 4C). As shown in Figures 4B and 4C, χ 3306 (ZK-kd) immunized mice showed significantly higher anti-LPS IgG compared to controls at more than 28 days after immunization ($p=0.007$). On the other hand, anti-LPS antibody levels in χ 3306 (ZK-k) immunized mice did not differ significantly from controls even 56 days after immunization, but a tendency to increase induction of the anti-LPS IgG was observed.

Page 10, line 35 – page 11, line 4:

Mice were intravenously vaccinated with triple-dose administration at 1×10^5 CFU/mouse of χ 3306 (ZK-k) or χ 3306 (ZK-kd). When total IgG titers were assayed, production was observed after two inoculations (Figure 4F). χ 3306 (ZK-kd) and χ 3306 (ZK-k) induced significantly more anti-LPS IgG than control at day 40 post-immunization ($p=0.017$ and $p=0.0005$, respectively).

Figure 4C and 4F:

Figure 4C and 4F have also been modified.

Page 12, line 12-14:

In comparison, all mice were survived in treatment with N61(ZK-kdo), formalin-killed N61(ZK-kdo) and saline (significant $p < 0.0001$).

Page 12, line 28 – page 13, line 5:

After a three-dose vaccination schedule, significant levels of total IgG against *E. coli* were present in all immunized mice on days 7 and 14 after vaccination with N61(ZK-kdo) compared to saline [significant $p < 0.05$; mean ODs in day 7, saline vs N61(ZK-kdo) = 0.15 vs 0.58; day 14, 0.19 vs 0.71], whereas no significant increase was detected in mice immunized with formalin-killed N61(ZK-kdo). On day 21 after vaccination, total IgG levels increased significantly in both N61(ZK-kdo) and formalin-killed N61(ZK-kdo) immunized mice [significant $p < 0.05$; mean ODs in day 21; saline vs formalin-killed N61(ZK-kdo) = 0.15 vs 0.52, saline vs N61(ZK-kdo) = 0.15 vs 0.78]. Notably, the live N61(ZK-kdo) was more immunogenic than the formalin-killed form [significant $p < 0.05$; mean ODs on day 7, formalin-killed N61(ZK-kdo) vs N61(ZK-kdo) = 0.21 vs 0.58; day 14, 0.31 vs 0.71; day 21, 0.52 vs 0.78].

Page 13, line 11-12:

In contrast, all vaccinated mice with N61(ZK-kdo) were alive [significant p = 0.0289 for formalin-killed N61(ZK-kdo) vs N61(ZK-kdo)].

□ “Mice were intraperitoneally vaccinated with a double-dose administration at 1×10^5 CFU/mouse of χ 3306 (ZK-k) or χ 3306 (ZK-kd).”

Justify the dosage use.

We thank for your pointing out. Correctly, mice were intravenously vaccinated with a triple-dose administration in the oral challenge test with *S. Typhimurium* χ 3306, as described above. This mistake in this paragraph and Fig. 4F have been revised (page 10, line 35 – page 11, line 1; page 33, Figure 4F).

□ Fig 4, panel arrangement can be improved, either from left to right or top to bottom.

In Figure 4, the relevant panels are indicated by distinct colors. Figure 4A is a safety test of vaccine candidates. Figures 4B-D and E-G show protection studies using intraperitoneal injection and oral infection models, respectively. As the reviewer points out, the order of the panels is not regular, but we believe the present layout is reasonable.

□ Fig 4 A, why the survival day was short (max 6 days)?

We modified Figure 4A to show survival rates through day 30 (page 33).

□ Fig 4 C and F, the standard unit for IgG titers should be IU/ml? Significant values and statistical tests for this experiment are not mentioned.

It is difficult to apply the standard unit, IU/ml, because the LPS was not a commercial product. The LPS used for the ELISA was extracted from *S. Typhimurium* χ 3306 using our LPS extraction kit.

To clarify the origin of the LPS, we have made an additional note in the section of Material and Methods as:

Page 21, line 30-35

“For ELISA, a 96-well plate (Nunc; Roskilde, Denmark) was coated with 50 μ l of 10 μ g/ml *S. Typhimurium* χ 3306 LPS extracted using LPS extraction kit (iNtRON Biotechnology, Inc.) and diluted in 50mM carbonate-bicarbonate buffer, pH 9.6. All incubations were carried out at 37°C for 60 min, and every incubation step was followed by four washes with ELISA wash buffer (0.9% NaCl supplemented with 0.1% Tween-20).”

Fig 5 C, histology stains are not clear. Suggest to show overview tissue stain, and high resolution of area that show bacterial aggregation.

We thank the reviewer for this constructive suggestion. We have modified Figure 5C to add the overview tissue stains, according to your suggestion (page 35, Fig 5C; page 36, line 5). In addition, to further clarify bacterial aggregation, images of immunostaining with *E. coli* antibodies have been added. As a result of immunostaining, local bacterial aggregation was confirmed in the dermal tissue in N61(ZK-kdo)-injected mice, which was not clear in HE stains. Moreover, we modified the following description:

page 12, in lines 18-21:

Tissue specimens of N61(ZK-kdo)-injected mice 24 hours after injection showed local bacteria aggregation in the dermal tissue. In contrast, bacteria aggregation in wild-type N61-injected mice was confirmed throughout the tissue. (Figure 5C).

Discussion:

Instead of using “quick and easy preparation”, suggest to use more precise, for example, the vaccine candidate can be prepared in how many days etc.

To indicate the specific details of the “quick and easy preparation”, we have made an additional note (page 14, line 6-7).

Suggest to include literature or evidence from previous studies to support the paragraph mentioned “The selection of the uAA incorporation system affected escape frequency.”

Following the reviewer's suggestion, we have made an additional note and cited additional references (page 14, line 9-14. Ref. 50,51).

Authors should discuss why the vaccine candidates only confer partial protection?

Although the only partial protection in survival rate was shown as the reviewer pointed out, the significant difference between the survival curves of the wild-type and the vaccine candidates indicates a protective effect of vaccination. In addition, *S. Typhimurium* commonly transmits via oral route (58). In our experiment, the vaccination exerted a better protective effect for survival rate against oral challenge than intraperitoneal challenge (Fig. 4D and G). This result suggests that our vaccination system has a potential to be put practical use.

We have summarized above remarks in the Discussion section as:

Page 17, line 8-11:

“Notably, the vaccination exerted a better protective effect for survival rate against oral challenge than intraperitoneal challenge (Fig. 4D and G). *S. Typhimurium* commonly transmits via oral route (58), suggesting that the uAA-auxotrophic suicide vaccines have a potential to be put practical use.”.

ref. 58; WHO (2018) “Salmonella (non-typhoidal)” Available at: [https://www.who.int/news-room/fact-sheets/detail/salmonella-\(non-typhoidal\)](https://www.who.int/news-room/fact-sheets/detail/salmonella-(non-typhoidal))

The limitation of study was not discussed.

The limitations of our research at this time have been discussed for each technology. In addition, we have summarized the technologies that are inadequately complete and added the direction of development required for translational research (page 18, line 6-14).

Materials and Methods:

Any reason for using young age (3-week-old) animal?

We thank for your pointing out. Correctly, 6-week-old mice were used for the experiment. The mistake in the Materials and Methods section was revised (page 20, line 19 and 30).

//

Response to Reviewer 3

This paper describes a novel approach for generating live-attenuated vaccines for bacterial pathogens using genetic engineering. Live attenuated vaccines are desirable because they are often more immunogenic and provide greater protection than other methods of vaccination; however, major problems can arise if the pathogen used as a vaccine is not effectively attenuated. In this paper, the authors used toxin-antitoxin genes to generate *E. coli* (laboratory and pathogenic strains) and *Salmonella enterica*, that can only survive in media supplemented with an unnatural amino acid (uAA). These engineered bacteria can be grown in media containing a uAA, which facilitates stop codon readthrough, and expression of an antitoxin gene. The cultured cells can then be used to inoculate an animal as a vaccine. Because the uAA does not exist within the animal, the antitoxin gene is not expressed, and the microbes die shortly after injection. Importantly, the bacteria are still able to illicit an immune response inducing the animal to generate antibodies to the pathogen. Building on their previous work, herein, the authors refine this method of generating bacteria that are auxotrophic for a uAA and successfully demonstrate the use of this technology for immunizing laboratory mice against pathogenic strains of bacteria.

This paper is well written and technically sound. The data are sufficient to support the conclusions made. I would rate the novelty of the work as moderate—uAAs have been used to generate live attenuated vaccines previously, as have toxin-antitoxin genes. Thus, this work is somewhat incremental. Moreover, it is limited to bacterial strains for which multiple expression plasmids are readily available. However, given the quality of the work and the pressing needs for rapid and effective methods for generating vaccines, I believe the work will be of interest to a diverse audience. I find no major issues that the authors must address at this time.

We thank the reviewers for their positive evaluation. The reviewers' comments are very encouraging to us as we conduct future research.

Re: Spectrum03557-23R1 (A novel vaccine strategy using quick and easy conversion of bacterial pathogens to unnatural amino acid-auxotrophic suicide derivatives)

Dear Dr. Yusuke Kato:

Your manuscript has been accepted, and I am forwarding it to the ASM production staff for publication. Your paper will first be checked to make sure all elements meet the technical requirements. ASM staff will contact you if anything needs to be revised before copyediting and production can begin. Otherwise, you will be notified when your proofs are ready to be viewed.

Sincerely,
Artem Rogovskyy
Editor
Microbiology Spectrum